# Bifidobacteria-Fermented Red Ginseng and Its Constituents Ginsenoside Rd and Protopanaxatriol Alleviate Anxiety/Depression in Mice by the Amelioration of Gut Dysbiosis

**DOI:** 10.3390/nu12040901

**Published:** 2020-03-26

**Authors:** Sang-Kap Han, Min-Kyung Joo, Jeon-Kyung Kim, Woonhee Jeung, Heerim Kang, Dong-Hyun Kim

**Affiliations:** 1Neurobiota Research Center, College of Pharmacy, Kyung Hee University, Seoul 02447, Korea; Koreask25h@naver.com (S.-K.H.); mkiti1727@gmail.com (M.-K.J.); kim_jk0225@naver.com (J.-K.K.); 2R&BD Center, Korea Yakult Co. Ltd., Yongin 17086, Korea; wjeung@re.yakult.co.kr (W.J.); heeerim@re.yakult.co.kr (H.K.)

**Keywords:** fermented red ginseng, ginsenoside Rd, protopanaxatriol, depression, gut microbiota

## Abstract

Gut dysbiosis is closely connected with the outbreak of psychiatric disorders with colitis. Bifidobacteria-fermented red ginseng (fRG) increases the absorption of ginsenoside Rd and protopanxatriol into the blood in volunteers and mice. fRG and Rd alleviates 2,4,6-trinitrobenzenesulfonic acid-induced colitis in mice. Therefore, to understand the gut microbiota-mediated mechanism of fRG against anxiety/depression, we examined the effects of red ginseng (RG), fRG, ginsenoside Rd, and protopanaxatriol on the occurrence of anxiety/depression, colitis, and gut dysbiosis in mice. Mice with anxiety/depression were prepared by being exposed to two stressors, immobilization stress (IS) or *Escherichia coli* (EC). Treatment with RG and fRG significantly mitigated the stress-induced anxiety/depression-like behaviors in elevated plus maze, light-dark transition, forced swimming (FST), and tail suspension tasks (TST) and reduced corticosterone levels in the blood. Their treatments also suppressed the stress-induced NF-κB activation and NF-κB^+^/Iba1^+^ cell population in the hippocampus, while the brain-derived neurotrophic factor (BDNF) expression and BDNF^+^/NeuN^+^ cell population were increased. Furthermore, treatment with RG or fRG suppressed the stress-induced colitis: they suppressed myeloperoxidase activity, NF-κB activation, and NF-κB^+^/CD11c^+^ cell population in the colon. In particular, fRG suppressed the EC-induced depression-like behaviors in FST and TST and colitis more strongly than RG. fRG treatment also significantly alleviated the EC-induced NF-κB^+^/Iba1^+^ cell population and EC-suppressed BDNF^+^/NeuN^+^ cell population in the hippocampus more strongly than RG. RG and fRG alleviated EC-induced gut dysbiosis: they increased Bacteroidetes population and decreased Proteobacteria population. Rd and protopanaxatriol also alleviated EC-induced anxiety/depression and colitis. In conclusion, fRG and its constituents Rd and protopanaxatriol mitigated anxiety/depression and colitis by regulating NF-κB-mediated BDNF expression and gut dysbiosis.

## 1. Introduction

Anxiety and depression disorders often co-occur both concurrently and sequentially [1,2]. Exposure to stressors such as immobilization and forced swimming stimulates the release of adrenal hormones in the adrenal gland and proinflammatory cytokines in the immune cells through the activation of hypothalamic-pituitary-adrenal (HPA) axis [3,4,5]. The excessive secretion of adrenal hormones and proinflammatory cytokines accelerates the occurrence of anxiety/depression by suppressing brain-derived neurotrophic factor (BDNF) expression in the brain, gut inflammation and dysbiosis by activating innate and adaptive immunities in the intestine [6,7,8]. Gut microbiota also modulates the immune and neural systems in the intestine via the microbiota-gut-brain (MGB) axis [9,10]. The overgrowth of *Escherichia coli* by immobilization stress (IS) and gavage of *Escherichia coli* cause gut dysbiosis, colitis, and anxiety in mice [11]. Gut dysbiosis is intimately connected with the systemic inflammation, resulting in obesity, autism, and depression [11,12]. Therefore, regulating gut dysbiosis can be important for the therapy of psychiatric disorders. 

Red ginseng (RG, the steamed root of *Panax ginseng* Meyer, family Araliaceae) is frequently used as a functional food and herbal medicine for inflammatory diseases, diabetes, and neuropsychiatric disorders [13,14,15]. Its well-known constituents are ginsenosides Rb1, Rg1, and Rg3 [16]. When RG is orally administered into humans or animals, its constituents, particularly ginsenosides, are metabolized into ginsenosides Rd, Rh2, compound K, and protopanaxatriol in the gastrointestinal tract. These metabolites are absorbed into the blood [17,18,19,20]. These metabolites have anti-inflammatory, anti-allergic, anti-tumor, and anti-diabetic effects [21,22]. Absorbable ginsenosides Rg1, Rg3, protopanaxadiol, and protopanaxtriol into the blood exhibit the potent anti-stress activity [23,24,25]. Therefore, to enforce the pharmacological activity of RG, absorbable ginsenoside-rich bifidobacteria-fermented RG (fRG) was developed [26,27,28]. fRG, which contains ginsenosides Rd, compound K (20-O-β-(D-Glucopyranosyl)-20(S)-protopanaxadiol), and protopanaxatriol more highly than RG [27,29], alleviates 2,4,6-trinitrobenzenesulfonic acid (TNBS)-induced colitis in mice more strongly than RG [27,28]. fRG and ginsenoside Rd alleviate the ovalbumin-induced allergic rhinitis in mice by the inhibition of IgE and IL-4 expression and modulation of gut microbiota composition [28]. However, the anti-psychiatric activities of RG and fRG still remain unclear.

Therefore, to understand the gut microbiota-mediated mechanism of RG and fRG against anxiety/depression, we examined the effects of RG, fRG, and their constituents ginsenoside Rd and protopanaxatriol on the immobilization stress (IS)- or *Escherichia coli* K1 (EC)-induced anxiety/depression, colitis, and gut dysbiosis in mice.

## 2. Materials and Methods 

### 2.1. Materials 

Antibodies for p-p65, p65, BDNF, cAMP response element-binding protein (CREB), p-CREB, and β-actin were purchased from Cell Signaling Technology (Beverly, MA). Enzyme-linked immunosorbent assay (ELISA) kit for corticosterone was purchased from Elabscience (Hebei, China). ELISA kits for TNF-α and IL-6 were purchased from eBioscience (San Diego, CA, USA). Fluoxetine hydrochloride (FL) was purchased from Alomone Labs (Jerusalem, Israel). Ginsenoside Rd and protopanaxatriol were purchased from Embo Laboratory (Daejeon, Korea). RG and fRG were prepared in Korea Yakult R&BD Center (Yongin, Korea), as previously reported [27,29]. 

### 2.2. Animals 

C57BL/6 mice (male, 19–21 g, 6 weeks old) were supplied from Koatech (Gyeonggi-do, Korea). Animals were fed with water and food ad libitum and maintained under the controlled condition (temperature, 22 °C ± 1 °C; humidity, 50% ± 10%; and light/dark cycle, 12 h). All animal experiments were approved by the Institutional Animal Care and Use Committee of the University (IACUC No KUASP(SE)-19-152) and performed according to the National Institute of Health (NIH) and University Guide for Laboratory Animal Care and Usage.

### 2.3. Preparation of Mice with Anxiety/Depression

Mice with anxiety/depression were prepared by exposure to IS or EC, as previously reported [7,30]. To decide the dosages of RG and fRG in the in vivo study, mice were randomly divided into eight groups (Con, Is, IR10, IR25, IL50, IF10, IF25, and IF50). Each group consisted of 6 mice. Each mouse (of Is, IR10, IR25, IL50, IF10, IF25, and IF50) except those in the control (Con) group was exposed to IS (12 h/day) for 2 days, as previously reported [7]. Thereafter, test agents (Con, saline [vehicle]; Is, vehicle; IR10, 10 mg/kg/day of RG; IR25, 25 mg/kg/day of RG; IR50, 50 mg/kg/day of RG; IF10, 10 mg/kg/day of fRG; IF25, 25 mg/kg/day of fRG; and IF50, 50 mg.kg/day of fRG, dissolved in 1% maltose) were orally gavaged once a day for 5 days and depressive behaviors were measured 24 h after the final treatment with test agents. Of these, treatment with RG or fRG at dosages of 10 and 25 mg/mouse/day significantly suppressed anxiety/depression (Appendix A). Therefore, we orally treated at dosages of 10 and 25 mg/mouse/day for the further in vivo study. 

First, to examine the effects of RG and fRG on IS-induced anxiety/depression, mice were randomly divided into seven groups (Con, Is, IPc, IRL, IPH, IFL, and IFH). Each group consisted of 6 mice. Each mouse (of Is, IPc, IRL, IPH, IFL, and IFH) except those in the control (Con) group was exposed to IS (12 h/day) for 2 days, as previously reported [7]. Thereafter, test agents (Con, saline [vehicle]; Is, vehicle; IPc, 10 mg/kg/day fluoxetine; IRL, 10 mg/kg/day of RG; IRH, 25 mg/kg/day of RG; IFL, 10 mg/kg/day of fRG; and IFH, 25 mg/kg/day of fRG, dissolved in 1% maltose) were orally gavaged once a day for 5 days and depressive behaviors were measured 24 h after the final treatment with test agents. 

Second, to examine the effects of RG and fRG on EC-induced anxiety/depression, mice were randomly divided into seven groups (Con, Ec, EPc, ERL, ERH, EFL, and EFH). Each group consisted of 6 mice. Each mouse (of Ec, EPc, ERL, ERH, EFL, and EFH) except those in the control (Con) group was orally gavaged with EC (1 × 10^9^ CFU/mouse/day, suspended in 0.2 mL saline) once a day for 5 days, as previously reported [11]. Thereafter, test agents (Con, saline [vehicle]; Ec, saline [vehicle]; EPc, 1 mg/kg/day buspirone; ERL, 10 mg/kg/day of RG; ERH, 25 mg/kg/day of RG; EFL, 10 mg/kg/day of fRG; and EFH, 25 mg/kg/day of fRG, dissolved in saline) were orally gavaged (for RG, fRG, and vehicle) or interaperitoneally injected (for buspirone) once a day for 5 days and depressive behaviors were measured 24 h after the final treatment with test agents. 

Third, to examine the effects of Rd and protopanaxatriol on EC-induced anxiety/depression, mice were randomly divided into four groups (Con, Ec, ERd, and EPt groups). Each group consisted of 6 mice. Each mouse (of Ec, ERd, and EPt groups) except those in the Con group was orally gavaged with EC as described above. Thereafter, test agents (Con, saline [vehicle]; Ec, vehicle; ERd, 5 mg/kg/day of ginsenoside Rd; EPt, 5 mg/kg/day of protopanaxatriol, dissolved in saline) were orally gavaged once a day for 5 day and depressive behaviors were measured 24 h after the final treatment with test agents. 

Mice were sacrificed 18 h after the final behavior tasks. Bloods, colons, and brains were then removed. The blood was centrifuged for the preparation of plasma fraction. Colon and brain hippocampal tissues were stored at −80 °C for further study.

### 2.4. Behavioral Tasks 

The elevated plus maze (EPM) task, light/dark transition test (LDT), tail suspension test (TST), and forced swimming test (FST) were performed as previously reported [11,30]. 

### 2.5. ELISA, Immunoblotting, Immunofluorescence Assay, and Myeloperoxidase Activity Assay

The immunofluorescence assay was performed in the sections of hippocampus and colon tissues as previously reported [11]. The immunoblotting was performed in the supernatants of hippocampus and colon homogenates as previously reported [11]. ELISA was assayed using ELISA kits as previously reported [11]. Myeloperoxiase activity was assayed in the colon supernatants as previously reported [7]. 

### 2.6. Fecal Microbita Composition Analysis 

The fresh stools of five mice (not trans-cardiacally perfused with 4% paraformaldehyde) were collected in the colon and bacterial genomic DNAs were extracted using a DNA isolation kit (QIAamp DNA stool mini kit), as previously reported [11,31]. Bacterial genomic DNAs were amplified using barcoded primers for the bacterial 16 S rRNA V4 region. Gene sequencing for each amplicon was carried out using Illumina iSeq 100 (San Diego, CA, USA). Prediction for functional genes was carried out using the phylogenetic investigation of communities by reconstruction of unobserved states (PICRUSt) [32,33]. Pyrosequencing reads have been deposited in the NCBI’s short read archive (PRJNA601857).

### 2.7. Statistical Analysis

All data are indicated as mean ± standard deviation (SD) and analyzed by Graph-Pad Prism 8 (GraphPad Software Inc., San Diego, CA, USA). The data were tested for normality before analysis of variance. The significance for data was analyzed by using one-way analysis of variance followed by Tukey’s multiple range test (as a post-hoc comparison). *p* values are indicated Appendix A.

## 3. Results

### 3.1. RG and FRG Alleviated the IS-Induced Anxiety/Depression in Mice

In order to understand the difference of the anti-anxiety/depression activity between RG and fRG, we examined the effects of RG and fRG on the IS-induced anxiety/depression in mice (Figure 1A–C). Exposure to IS increased anxiety/depression-like behaviors in the EPM task, LDT, and FST. However, oral administration of RG or fRG significantly alleviated the IS-induced anxiety/depression-like behaviors. RG or fRG treatment increased the IS-suppressed time spent in the open arms (OT) and open arm entries (OE) in the EPM task. RG and fRG mitigated the IS-induced anxiety/depression-like behaviors in LDT and FST. fRG at a dose of 25 mg/kg/day (fRG-H) alleviated depressive behaviors in the FST more strongly than that of RG at a dose of 25 mg/kg/day (RG-H). Treatment with RG or fRG also suppressed the IS-induced NF-κB activation and increased BDNF expression in the hippocampus (Figure 1D). Furthermore, oral administration of RG or fRG suppressed IS-induced corticosterone and IL-6 expression in the blood (Figure 1E,F).

Next, we examined the effects of RG and fRG on the IS-induced colitis in mice. IS exposure-induced colitis caused colon shortening and increased myeloperoxidase activity, TNF-α and IL-6 expression, and NF-κB activation in the colon (Figure 1G–K). However, oral administration of RG or fRG suppressed the IS-induced myeloperoxidase activity and NF-κB activation in the colon. fRG-H most potently suppressed the IS-induced TNF-α expression in the colon, followed by RG-H.

### 3.2. RG and FRG Alleviated EC-Induced Anxiety/Depression, Colitis, and Gut Dysbiosis in Mice

Next, we examined the effects of RG and fRG on the EC-induced anxiety/depression in mice. Exposure to EC increased anxiety/depression-like behaviors in EPM, LDT, TST, and FST (Figure 2A–D). However, treatment with RG or fRG significantly alleviated the EC-induced anxiety/depression-like behaviors. Oral gavage of RG or fRG increased EC-suppressed OT and OE in the EPM task: at a dose of 25 mg/kg/day they increased OT to 54.8% and 65.6% of the control group, respectively. They also suppressed the EC-induced anxiety/depression-like behaviors in LDT, FST and TST. fRG-H most strongly suppressed depressive behaviors in FST and TST, followed by fRG at a dose of 10 mg/kg/day (fRG-L) and RG-H. Oral administration of RG or fRG also suppressed the EC-induced corticosterone and IL-6 expression (Figure 2E,F). Of tested fRG and RG, fRG-H most strongly suppressed the EC-induced corticosterone levels in the blood. Treatment of RG or fRG suppressed EC-induced NF-κB activation and NF-κB^+^/Iba1^+^ cell population in the hippocampus, while the BDNF expression, CREB phosphorylation, and BDNF^+^/NeuN^+^ cell population were induced (Figure 2G–I, Appendix A). Although the induction of BDNF^+^/NeuN^+^ cell population by fRG-H was not significantly different to that by RG-H, fRG-L increased EC-suppressed BDNF^+^/NeuN^+^ cell population more strongly than did RG at a dose of 10 mg/kg/day (RG-L). fRG-H and fRG-L significantly suppressed EC-induced NF-κB^+^/Iba1^+^ cell population more strongly than RG-H and RG-L, respectively.

We also examined the effects of RG and fRG on the EC-induced colitis in mice. Exposure to EC also induced colitis in mice, causing colon shortening and increased myeloperoxidase activity, TNF-α and IL-6 expression, NF-κB activation, and infiltration of NF-κB/CD11c+ cells (activated DCs and macrophages) in the colon (Figure 3A–F). However, oral gavage of RG or fRG suppressed EC-induced myeloperoxidase activity, NF-κB activation, and NF-κB/CD11c+ cell population in the colon. Of tested fRG and RG, treatment with fRG-H most strongly suppressed EC-induced colon shortening, myeloperoxidase activity, TNF-α and IL-6 expression, and NF-κB activation in the colon. 

Next, we examined the effects of RG and fRG on the gut microbiota composition in mice with EC-induced anxiety/depression. Exposure to EC did not affect the bacterial richness and α-diversity, as demonstrated by the estimated operational taxonomic unit (OTU) richness and Shannon’s diversity index while the β-diversity was changed by using the performing the principal coordinate analysis (Figure 3G–I). Treatment with RG or fRG modulated EC-induced β-diversity change, while the α-diversity was not affected. The fecal bacterial community of EC-exposed mice was significantly different from that of control mice. The Proteobacteria and Firmicutes populations showed a higher abundance in the EC-exposed mice than in control mice while the Bacteroidetes population showed a lower abundance. However, treatment with RG or fRG suppressed the EC-induced abundance of Proteobacteria and Firmicutes populations and increased the EC-suppressed abundance of Bacteroidetes population (Figure 3J,K, Appendix A, Appendix A). Treatment with RG increased the EC-suppressed Odoridaceae, Bacteroidaceae, and Rikenellaceae populations in the family level and Odoribacter, PAC001074_g, Bacteroides, and Alisptipes populations in the genus level. Treatment with fRG increased the EC-suppressed abundance of Bacteroidaceae and Muribaculaceae populations in the family level and Alistipes, Bacteroides, Paraprevotella, and PAC001704_g populations in the genus level. However, treatment with RG or fRG highly, but not significantly, reduced the EC-induced Enterobacteriaceae, Lachnospiraceae, and Ruminococcaceae populations and Escherichia, Oscillibacter populations in the genus level.

### 3.3. Ginsenoside Rd and Protopanaxatriol Alleviated EC-Induced Anxiety/Depression, Colitis, and Gut Dysbiosis in Mice

The effects of Rd and protopanaxatriol on the occurrence of depression and colitis were examined in mice with EC-induced anxiety/depression. Oral administration of Rd or protopanaxatriol significantly alleviated the EC-induced anxiety/depression-like behaviors (Figure 4A–D). Treatment with Rd alleviated anxiety/depression-like behaviors in the EPM task, while it was not alleviated by treatment with protopanaxatriol. However, they suppressed the EC-induced anxiety/depression-like behaviors in LDT, TST, and FST. Furthermore, Rd and protopanaxatriol suppressed the EC-induced corticosterone and IL-6 expression in the blood (Figure 4E,F). They suppressed the EC-induced NF-κB activation and NF-κB^+^/Iba1^+^ cell population and increased the EC-suppressed BDNF expression, CREB phosphorylation, and BDNF^+^/NeuN^+^ cell population in the hippocampus (Figure 4G–I, Appendix A).

Oral gavage of Rd or protopanaxtriol also suppressed EC-induced colitis: they inhibited myeloperoxidase activity, NF-κB activation, and NF-κB^+^/CD11c^+^ cell population in the colon (Figure 5A–F). Rd treatment alleviated the EC-induced anxiety/depression and colitis more potently than protopanaxatriol treatment. We also examined the effects of Rd and protopanaxatriol on the gut microbiota composition in mice with EC-induced anxiety/depression (Figure 5G–K, Appendix A, Appendix A). Exposure to EC did not affect the α-diversity while the β-diversity was changed. Treatment with Rd or protopanaxatriol suppressed the shiftness of β-diversity by EC while the α-diversity was not affected. At the phylum level, the EC-exposed mice showed a higher abundance in the Proteobacteria and Firmicutes populations while the Bacteroidetes population showed a lower abundance. Treatment with Rd increased the EC-suppressed Odoribacteraceae and Helicobacteriaceae populations in the family level and Oridobacter, Helicobacter, and PAC001124_g populations in the genus level. Treatment with protopanaxtriol increased the EC-suppressed Bacteroidaceae population in the family level and Bacteroides, Odoribacter, and Paraprevotella populations in the genus level. However, treatment with Rd or protopanaxatriol suppressed the EC-induced Enterobacteriaceae population and Escherichia population in the genus level.

## 4. Discussion

Exposure to stressors such as immobilization and antibiotics causes anxiety/depression as well as gut dysbiosis through the activation of the HPA axis [5,33]. Exposure to immobilization stress caused anxiety/depression in germ-free animals more exaggeratedly than it did in specific pathogen-free ones [34]. The fecal transplantation of conventional mice alleviates the exaggeratedly activated anxiety in germ-free mice [34,35]. Exposure with ampicillin also causes anxiety and gut dysbiosis in conventional mice [36]. However, the transplantation of conventional rat feces alleviates antibiotics-induced anxiety and gut dysbiosis in the transplanted rats [37]. These findings suggest that the induction of gut dysbiosis by exposure to stressors can cause psychiatric disorders such as anxiety/depression. 

In the present study, we also found that exposure to IS or EC caused anxiety/depression in mice, as previously reported [11,30]. They induced the NF-κB activation, IL-6 expression, and NF-κB^+^/Iba1^+^ cell population in the hippocampus and corticosterone and IL-6 levels in the blood, while the BDNF expression was suppressed in the hippocampus. However, RG and fRG suppressed IS- or EC-induced anxiety/depressive behaviors, NF-κB activation, and NF-κB^+^/Iba1^+^ cell population and increased the BDNF expression and BDNF^+^/NeuN^+^ cell population in the hippocampus. Kim et al. reported that acetate-treated RG alleviated the forced swimming-induced depression in mice in FST [38]. Baek et al. reported that RG might stabilize the sympathetic nervous system in individuals with high stress [39]. Tode et al. reported that RG alleviated fatigue, insomnia, and depression in patients with severe climacteric syndromes [40]. These results suggest that RG and fRG can alleviate anxiety/depression by the regulation of NF-κB-mediated BDNF expression. RG and fRG also alleviated the EC- or IS-induced colitis in mice; they suppressed the EC-induced myeloperoxidase activity, NF-κB activation, and TNF-α and IL-6 expression, and NF-κB^+^/CD11c^+^ cell population in the colon. These results suggest that RG and fRG can simultaneously alleviate anxiety/depression and colitis by suppressing NF-kB activation in the gut and brain. Of these, fRG more strongly, but not significantly, alleviated IS-induced depression-like behaviors in FST and TNF-α expression in the colon than RG. fRG also alleviated EC-induced depression-like behaviors in FST, TST, and NF-κB^+^/Iba1^+^ cell populations and EC-suppressed BDNF^+^/NeuN^+^ cell population in the hippocampus more strongly than RG. Furthermore, fRG also alleviated colitis in mice more strongly than RG. These results suggest that fRG can alleviate depression and colitis more strongly than RG. 

We also found that exposure to EC caused gut dysbiosis; the Proteobacteria and Firmicutes populations showed a higher abundance while the Bacteroidetes population showed a lower abundance. Jang et al. reported that exposure to IS induced colitis and gut dysbiosis in mice. They also found that exposure to IS increased *Escherichia coli* population in the gut microbiota of mice and oral gavage of IS-inducible *Escherichia coli* caused colitis and anxiety in mice [11]. In the present study, we also found that exposure to IS caused colitis and anxiety in mice and exposure to *Escherichia coli* K1 caused colitis and anxiety/depression in mice. Kim et al. reported that fRG alleviate gut dysbiosisa in mice with ovalbumin-induced allergic rhinitis [28]. These results support the suggestion that the induction of gut dysbiosis by the exposure to stressors can cause colitis. Oral administration of RG or fRG significantly alleviated EC-induced gut dysbiosis in mice; these suppressed the EC-induced abundance of the Proteobacteria and Firmicutes populations. They also alleviated the EC- or IS-induced colitis in mice as they suppressed the EC-induced myeloperoxidase activity, NF-κB activation, TNF-α and IL-6 expression, and NF-κB^+^/CD11c^+^ cell population in the colon. These results suggest that RG and fRG can alleviate colitis as well as psychiatric disorders by the amelioration of gut dysbiosis. 

In the present study, orally administered RG and fRG alleviated EC- or IS-induced anxiety/depression in mice. In addition, many constituents of orally administered RG and fRG are not easily absorbed from the intestine into the blood [16]. Therefore, these constituents contact with gut microbiota in the gastrointestinal tract and are metabolized to easily absorbable compounds such as compound K. When RG and fRG are orally administered in humans and mice, the ginsenosides most highly absorbed into the blood were Rd followed by protopanaxatriol and compound K [27,29], Appendix A. In the present study, we found that oral administration of ginsenoside Rd and protopanaxatriol significantly suppressed the IS- or EC-induced depression-like behaviors, NF-κB activation, and NF-κB^+^/Iba1^+^ cell population in the hippocampus of mice, while the BDNF expression and CREB phosphorylation increased. They also reduced the EC-induced corticosterone and IL-6 levels in the blood. Furthermore, they alleviated IS- or EC-induced colitis; they inhibited NF-κB activation and NF-κB^+^/CD11c^+^ cell population in the colon. RG and fRG also alleviated the EC-induced Proteobacteria populations in the gut microbiota. fRG and ginsenoside Rd alleviate gut dysbiosis in mice with ovalbumin-induced allergic rhinitis [28]. These results suggest that fRG, RG, and their ginsenosides, particularly ginsenoside Rd, can alleviate anxiety/depression and colitis by modulating the NF-κB-mediated BDNF expression and gut dysbiosis. Moreover, ginsenosides Rb1, Rg1, Rg2, Rg3, Rh2, Re, and compound K mitigate depression in rodents [14,23,24,41,42,43,44,45,46]. These findings suggest that the anti-depressive effects of RG and fRG can be dependent on the absorption of these ginsenosides including ginsensides Rd and Rg1 and protopanaxatriol into the blood. Kim et al. reported that ginsenosides Rd, protopanaxatriol, and compound K are more highly absorbed into the blood of volunteers and mice orally administered with fRG compared to those orally treated with RG, while the absorption of ginsenoside Rg1 into the blood was not different [27,29], Appendix A. These findings suggest that the anti-anxiety/depressive effect of fRG was more potent than that of RG and may be due to the high absorption of ginsenosides, particularly ginsenoside Rd and protopanaxtriol, and the potent attenuation of gut and hippocampal inflammation via the regulation of gut dysbiosis. 

## 5. Conclusions

The induction of gut dysbiosis by exposure to stressors IS and *Escherichia coli* K1 can cause anxiety/depression with colitis, while the potent alleviation of stress-induced gut dysbiosis and inflammation by fRG may be helpful in alleviating anxiety/depression through the amelioration of gut and hippocampal inflammation.

## Figures and Tables

**Figure 1 nutrients-12-00901-f001:**
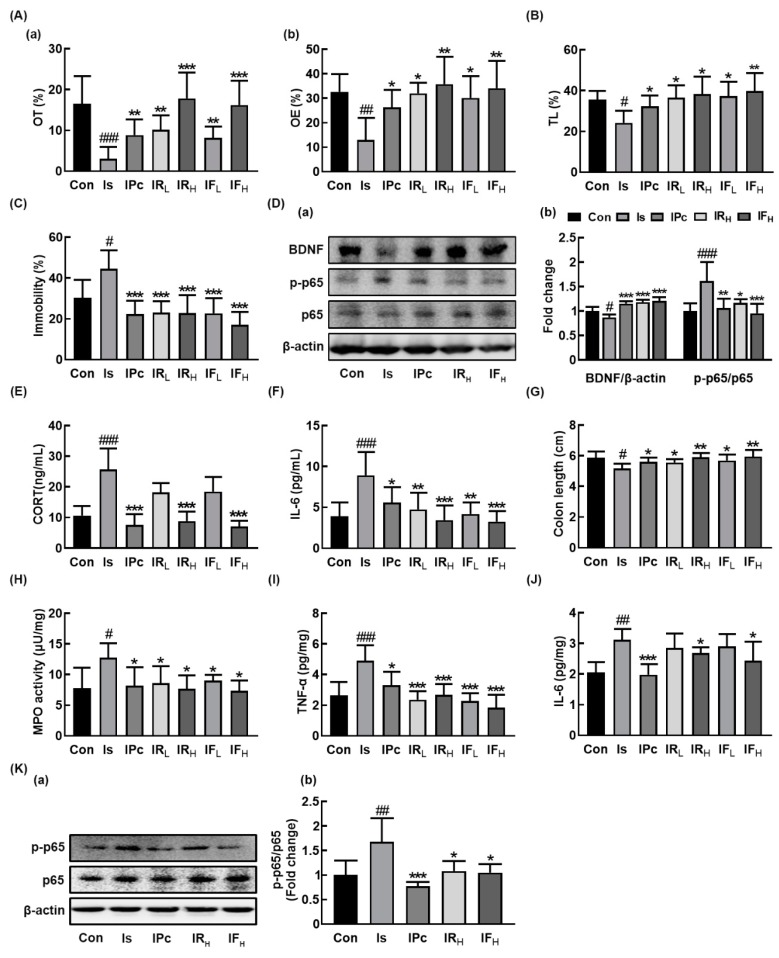
Oral administration of RG or fRG alleviated immobilization stress (IS)-induced anxiety/depression and colitis in mice. Effects on anxiety/depression-like behaviors in elevated plus maze (**A**: (**a**), open arms (OT); (**b**), open arm enteries (OE), light/dark transition test (**B**), and forced swimming test (**C**). (**D**) Effects on brain-derived neurotrophic factor (BDNF) expression, NF-κB activation (**a**), and their intensities of BDNF/β-actin and p-p65/p65 (**b**) in the hippocampus. Effects on corticosterone (CORT, **E**) and IL-6 levels (**F**) in the blood. Effects on colon length (**G**), myeloperoxidase (MPO) activity (**H**), TNF-α (**I**) and IL-6 (**J**) expression, and NF-κB activation (**K**(**a**)) and its intensity (K(**b**)) in the colon. Mice were exposed to immobilization stress (IS) and test agents (Is, vehicle [saline]; IPc, 10 mg/kg/day of fluoxetine; IRL, 10 mg/kg/day of RG; IRH, 25 mg/kg/day of RG; IFL, 10 mg/kg/day of fRG; IFH, 25 mg/kg/day of fRG) were gavaged daily for five days. Normal control group (Con) not exposed to IS was treated with saline instead of test agents. Hippocampal p65, p-p65, BDNF, and β-actin were analyzed by immunoblotting. Blood IL-6 and corticosterone levels were assayed by ELISA kits. Data values were indicated as mean ± SD (*n* = 6). ^#^
*p* < 0.05 vs. Con group. ^##^
*p* < 0.01 vs. Con group. ^###^
*p* < 0.001 vs. Con group. * *p* < 0.05 vs. Is group. ** *p* < 0.01 vs. Is group. *** *p* < 0.001 vs. Is group.

**Figure 2 nutrients-12-00901-f002:**
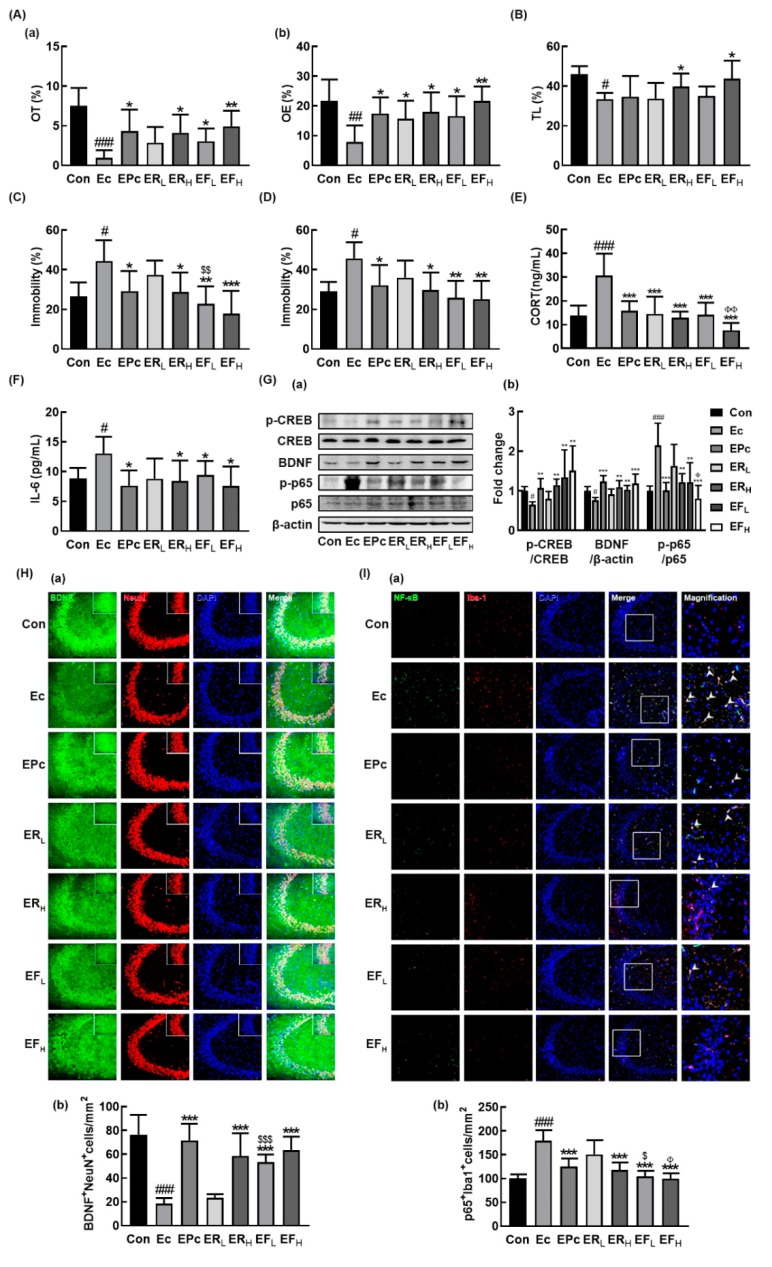
Oral administration of RG or fRG alleviated *Escherichia coli* K1 (EC)-induced anxiety/depression and colitis in mice. Effects on anxiety/depression-like behaviors in elevated plus maze (**A**: (**a**), OT; (**b**), OE), light/dark transition test (**B**), forced swimming test (**C**), and tail suspension test (**D**). Effects on corticosterone (CORT, **E**) and IL-6 levels (**F**) in the blood. (**G**) Effects on BDNF expression, cAMP response element-binding protein (CREB) phosphorylation, and NF-κB activation (**a**) and their intensities of (**b**) in the hippocampus. Effects on the infiltration of BDNF^+^/NeuN^+^ (H(**a**)) and NF-κB^+^/Iba1^+^ cells (I(**a**)) into the CA3 region of hippocampus and counts of BDNF^+^/NeuN^+^ cells (H(b)) and NF-κB^+^/Iba1^+^ cells (I(b)). Mice were exposed to EC (1×10^9^ CFU/mouse/day) and test agents (Ec, vehicle [saline]; EPc, 1 mg/kg/day of buspirone; ERL, 10 mg/kg/day of RG; ERH, 25 mg/kg/day of RG; EFL, 10 mg/kg/day of fRG; EFH, 25 mg/kg/day of fRG) were gavaged (for vehicle, RG, and fRG) or intraperitoneally injected (for buspirone) daily for five days. Normal control group (Con), not exposed to EC, was treated with saline instead of test agents. p65, p-p65, CREB, p-CREB, and β-actin were analyzed by immunoblotting. Blood IL-6 and corticosterone levels were assayed by ELISA kits. Data values were indicated as mean ± SD (*n* = 6)). ^#^
*p* < 0.05 vs. Con group. ^##^
*p* < 0.01 vs. Con group. ^###^
*p* < 0.001 vs. Con group. * *p* < 0.05 vs. Ec group. ** *p* < 0.01 vs. Ec group. *** *p* < 0.001 vs. Ec group. ^$^
*p* < 0.05 vs. ER_L_ group. ^$$^
*p* < 0.01 vs. ER_L_ group. ^$$$^
*p* < 0.001 vs. ER_L_ group. ^Φ^
*p* < 0.05 vs. ER_H_ group. ^ΦΦ^
*p* < 0.01 vs. ER_H_ group.

**Figure 3 nutrients-12-00901-f003:**
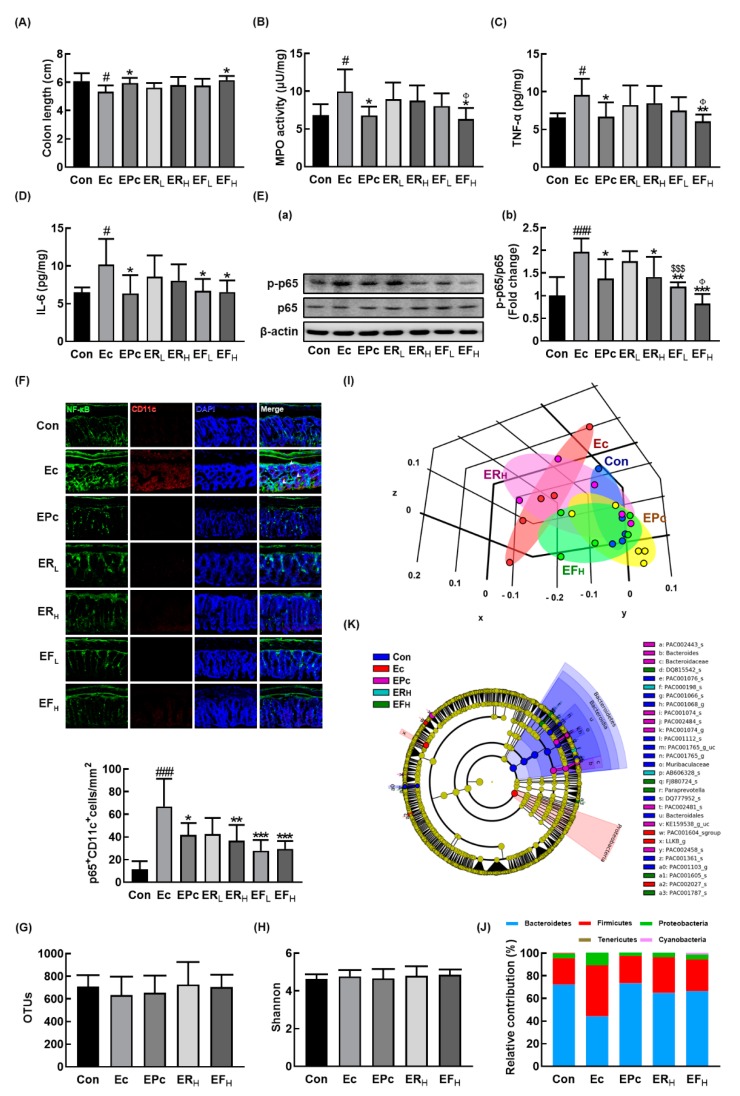
Oral administration of RG or fRG alleviated *Escherichia coli* K1 (EC)-induced colitis and gut dysbiosis in mice. Effects on colon length (**A**), myeloperoxidase (MPO) activity (**B**), TNF-α (**C**), IL-6 (**D**), NF-κB activation (**E**(**a**) and its intensity (**E**(**b**)), and NF-κB^+^/CD11c^+^ cells (**F**) in the colon. Effect on the composition of gut microbiota, analyzed by the pyrosequencing: operational taxonomic unit (OTUs) (**G**), Shannon (**H**), principal coordinate analysis (PCoA) plot (**I**) based on Jensen-Shannon analysis, phylum (**J**), and cladogram (**K**) generated by LEfSE indicating significant differences in gut microbial abundances among NC (blue), Ec (red), EPc (purple), ERH (blue-green), and EFH (green) groups. The threshold logarithmic score was set at 3.5 and ranked. Yellow nodes represent species with no significant difference. Mice were treated with EC and test agents as described in Figure 2. p65, p-p65, and β-actin were analyzed by immunoblotting. Blood IL-6 and corticosterone levels were assayed by ELISA kits. Data values were indicated as mean ± SD (*n* = 6). ^#^
*p* < 0.05 vs. Con group. ^###^
*p* < 0.001 vs. Con group. * *p* < 0.05 vs. Ec group. ** *p* < 0.01 vs. Ec group. *** *p* < 0.001 vs. Ec group. ^$$$^
*p* < 0.001 vs. ER_L_ group. ^Φ^
*p* < 0.05 vs. ER_H_ group.

**Figure 4 nutrients-12-00901-f004:**
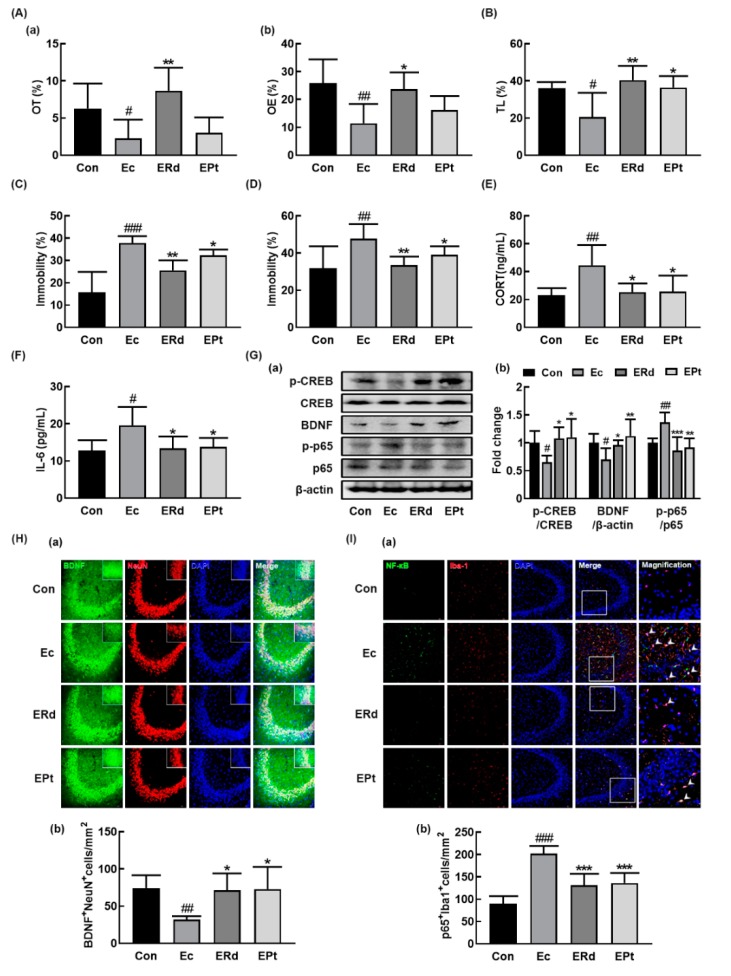
Oral administration of ginsenoside Rd and protopanaxatriol alleviated *Escherichia coli* K1 (EC)-induced anxiety/depression and colitis in mice. Effects on anxiety/depression-like behaviors in elevated plus maze (**A**: (**a**), OT; (**b**), OE), light/dark transition test (**B**), forced swimming test (**C**), and tail suspension test (**D**). Effects on corticosterone (CORT, **E**) and IL-6 levels (**F**) in the blood. (**G**) Effects on BDNF expression, CREB phosphorylation, and NF-κB activation in the hippocampus. Effects on the infiltration of BDNF^+^/NeuN^+^ (**H**) and NF-κB^+^/Iba1^+^ cells (**I**) into the CA3 region of hippocampus. Mice were exposed to EC (1 × 10^9^ CFU/mouse/day) and test agents (Ec, vehicle [saline]; ERd, 5 mg/kg/day of ginsenoside Rd; EPt, 5 mg/kg/day of protopanaxatriol) were gavaged daily for 5 days. Normal control group (Con), not exposed to EC, was treated with saline instead of test agents. p65, p-p65, CREB, p-CREB, and β-actin were analyzed by immunoblotting. Blood IL-6 and corticosterone levels were assayed by ELISA kits. Data values were indicated as mean ± SD (*n* = 6)). ^#^
*p* < 0.05 vs. Con group. ^##^
*p* < 0.01 vs. Con group. ^###^
*p* < 0.001 vs. Con group. * *p* < 0.05 vs. Ec group. ** *p* < 0.01 vs. Ec group. *** *p* < 0.001 vs. Ec group. ^$^
*p* < 0.05 vs. ER_L_ group.

**Figure 5 nutrients-12-00901-f005:**
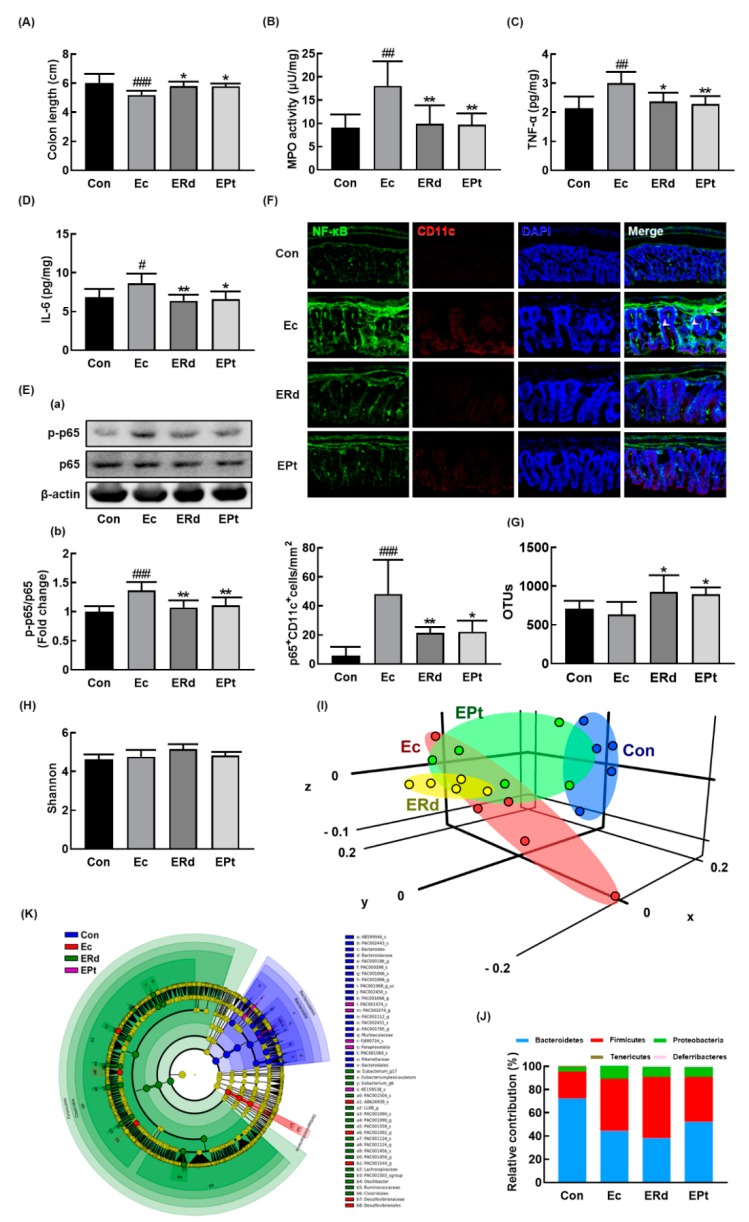
Oral administration of Rd or protopanaxatriol alleviated *Escherichia coli* K1 (EC)-induced colitis and gut dysbiosis in mice. Effects on colon length (**A**), myeloperoxidase (MPO) activity (**B**), TNF-α (**C**), IL-6 (**D**), NF-κB activation (**E**(**a**)) and its intensity (**E**(**b**)), and NF-κB+/CD11c+ cells (**F**) in the colon. Effect on the composition of gut microbiota analyzed by the pyrosequencing: OTUs (**G**), Shannon (**H**), principal coordinate analysis (PCoA) plot (**I**) based on Jensen-Shannon analysis, pyrlum (**J**), and cladogram (**K**) generated by LEfSE indicating significant differences in gut microbial abundances among NC (blue), Ec (red), ERd (green), and EPt (purple) groups. The threshold logarithmic score was set at 3.5 and ranked. Yellow nodes represent species with no significant difference. Con and EC mouse microbiota composition data were used in those of Figure 3. Mice were treated with EC and test agents, as described in Figure 4. p65, p-p65, and β-actin were analyzed by immunoblotting. Blood IL-6 and corticosterone levels were assayed by ELISA kits. Data values were indicated as mean ± SD (*n* = 6). # *p* < 0.05 vs. Con group. * *p* < 0.05 vs. EC group). ^#^
*p* < 0.05 vs. Con group. ^##^
*p* < 0.01 vs. Con group. ^###^
*p* < 0.001 vs. Con group. * *p* < 0.05 vs. Ec group. ** *p* < 0.01 vs. Ec group.

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
