# Peer review of "Bifidobacteria-Fermented Red Ginseng and Its Constituents Ginsenoside Rd and Protopanaxatriol Alleviate Anxiety/Depression in Mice by the Amelioration of Gut Dysbiosis"

_nutrients, 2020, doi:10.3390/nu12040901_

Round 1

Reviewer 1 Report

In this article, the authors demonstrated that bifidobacteria-fermented red ginseng (fRG) and its constituents ginsenoside Rd and protopanaxatriol are able to alleviate anxiety/depression behavior. The authors use both immobilization stress (IS) and E. coli (EC) induced mice anxiety/depression model to evaluate the potential therapeutic role of RG and fRG. They found that RG and fRG significantly mitigated stress-induced anxiety/depression behavior by the amelioration of gut dysbiosis. The authors present the data in a very convincing manner.

I still have one concern regarding the authors’ statement in line 142 “Overall, fRG alleviated IS-induced anxiety/depression more potently than RG” as well as the overall conclusion in line 30 that “fRG more potently alleviated anxiety/depression and colitis than RG”. In Fig. 1 and 2, both RG and fRG treatment have significant effect comparing with untreated group, suggesting that RG and fRG both alleviate IS and EC induced anxiety/depression. More evidence needs to be provided (such as statistically analysis between RG and fRG group) to support the notion of fRG is more potently alleviate the anxiety/depression and colitis thatn RG.

Author Response

Comments and Suggestions for Authors

In this article, the authors demonstrated that bifidobacteria-fermented red ginseng (fRG) and its constituents ginsenoside Rd and protopanaxatriol are able to alleviate anxiety/depression behavior. The authors use both immobilization stress (IS) and E. coli (EC) induced mice anxiety/depression model to evaluate the potential therapeutic role of RG and fRG. They found that RG and fRG significantly mitigated stress-induced anxiety/depression behavior by the amelioration of gut dysbiosis. The authors present the data in a very convincing manner.

I still have one concern regarding the authors’ statement in line 142 “Overall, fRG alleviated IS-induced anxiety/depression more potently than RG” as well as the overall conclusion in line 30 that “fRG more potently alleviated anxiety/depression and colitis than RG”.

-->Thank you for your comments. We revised it (we added the detailed description).

In Fig. 1 and 2, both RG and fRG treatment have significant effect comparing with untreated group, suggesting that RG and fRG both alleviate IS and EC induced anxiety/depression. More evidence needs to be provided (such as statistically analysis between RG and fRG group) to support the notion of fRG is more potently alleviate the anxiety/depression and colitis thatn RG

-->Thank you. We revised all figures according to your comment. And we added the detailed description.

Reviewer 2 Report

Comments to the Authors:

The manuscript deals with an interesting subject. The potential link between gut dysbiosis and anxiety/depression symptoms, together with the putative positive effects of red ginseng and Bifidobacteria-red ginseng to alleviate those symptoms, are relevant in this field.

The introduction is clear and the experimental design looks correct. Nonetheless, the presentation of results and discussion are confusing and should be substantially revised. Specific comments are given below.  

In the Material and methods section there are a number of differents points to be addressed. How were collected the samples for microbial community studies? Further details of the analysis of the gut microbiota composition should be given. The statistical analysis is not clearly described. Which were the fixed and random effects in the statistical analysis? Why did you choose the Duncan multiple range test as the post-hoc test? You made some specific comparisons and they should be explained.

Along the text you state that the effect of fRG to alleviate anxiety/depression or colitis is stronger than RG, although you do not show statistical differences among the different treatments (e.g., among IPc, IRL, IRH, IFL, IFH). You need to support your statement with statistical results.

The bar graphs are reasonably clear (except the statistical difference among treatments as previously mentioned), although the other graphs (e.g., 1D, 1K, 2H and 2I) are very difficult to be understood.

You state that “β-diversity was increased”. You present this β-diversity through a principal coordinate analysis, so I do not understand what you mean with this sentence. Please, explain further and clearer this issue.

The results on the microbial composition are very confusing. In the presented figures it is not possible to see the significant difference in abundances (relative contirbutions) and the populations that apparently increased with RG or fRG (as described in the text) do not always appear in the supplemental Tables or the results presented there do not match with those explained in the text. The description of results on the microbial composition should be substantially improved.

Unlike stated in the text, Figures 4Aa and b do not show an effect of protopanaxatriol in the EPM task.

The discussion should be revised according with the modifications suggested in the results, especially with regards to the effects on the gut microbial composition.

Other comments:

You briefly mention that a preliminary study was conducted to select the dosages of fRG and RG and you should give more details of this assay (e.g., duration, number of animals, ….).

In the first sentence of results you seem to refer to a published study. That should then be included in the discussion and not here, and if the data shown in Table S1 comes from a previously published study, they should not be reported here but only refer to them. Again, in the first sentence of the section 3.2 you cite a published study and references should be used only in the discussion.

Lines 224-234: The letter size is different to the rest of the text and in the first line there is an error because I assume you refer to the “oral gavage of Rd and protopanaxatriol” and not “of RG and fRG”.

Please, revise the word “protopanaxatriol” along the text because it is not always well written.

The English writing should be revised to improve the understanding of the manuscript.

Author Response

Comments to the Authors:

The manuscript deals with an interesting subject. The potential link between gut dysbiosis and anxiety/depression symptoms, together with the putative positive effects of red ginseng and Bifidobacteria-red ginseng to alleviate those symptoms, are relevant in this field.

The introduction is clear and the experimental design looks correct. Nonetheless, the presentation of results and discussion are confusing and should be substantially revised. Specific comments are given below.

--> Thank you. Our manuscript was revised according to your comment.

In the Material and methods section there are a number of differents points to be addressed. How were collected the samples for microbial community studies? Further details of the analysis of the gut microbiota composition should be given.

--> Thank you for your comment. We added the method 2.5. for the microbiota composition sequencing in Materials and Methods section.

The statistical analysis is not clearly described. Which were the fixed and random effects in the statistical analysis? Why did you choose the Duncan multiple range test as the post-hoc test? You made some specific comparisons and they should be explained.

--> Thank you for your comment. We hope to simultaneously analyze (1) the effectiveness of RG and fRG and the difference between the effectiveness of RG and fRG.

Along the text you state that the effect of fRG to alleviate anxiety/depression or colitis is stronger than RG, although you do not show statistical differences among the different treatments (e.g., among IPc, IRL, IRH, IFL, IFH). You need to support your statement with statistical results.

-->Thank you for your comment. We revised all figures and added its related description.

The bar graphs are reasonably clear (except the statistical difference among treatments as previously mentioned), although the other graphs (e.g., 1D, 1K, 2H and 2I) are very difficult to be understood.

-->Thank you for your comment. We described it in detail in figures 1 and 2 legends.

You state that “β-diversity was increased”. You present this β-diversity through a principal coordinate analysis, so I do not understand what you mean with this sentence. Please, explain further and clearer this issue.

-->Thank you for your comment. We revised it.

The results on the microbial composition are very confusing. In the presented figures it is not possible to see the significant difference in abundances (relative contirbutions) and the populations that apparently increased with RG or fRG (as described in the text) do not always appear in the supplemental Tables or the results presented there do not match with those explained in the text. The description of results on the microbial composition should be substantially improved.

-->Thank you for your comment. We revised it.

Unlike stated in the text, Figures 4Aa and b do not show an effect of protopanaxatriol in the EPM task.

-->Thank you for your comment. We revised it.

The discussion should be revised according with the modifications suggested in the results, especially with regards to the effects on the gut microbial composition.

-->Thank you. We significantly changed the discussion section according to your comment.

Other comments:

You briefly mention that a preliminary study was conducted to select the dosages of fRG and RG and you should give more details of this assay (e.g., duration, number of animals, ….).

-->Thank you. We added the content in Materials and Methods section according to your comment.

In the first sentence of results you seem to refer to a published study. That should then be included in the discussion and not here, and if the data shown in Table S1 comes from a previously published study, they should not be reported here but only refer to them. Again, in the first sentence of the section 3.2 you cite a published study and references should be used only in the discussion.

-->Thank you for your comment. We revised it.

Lines 224-234: The letter size is different to the rest of the text and in the first line there is an error because I assume you refer to the “oral gavage of Rd and protopanaxatriol” and not “of RG and fRG”.

-->Thank you for your comment. We revised it.

Please, revise the word “protopanaxatriol” along the text because it is not always well written.

-->Thank you for your comment. We revised it.

The English writing should be revised to improve the understanding of the manuscript.

-->Thank you for your comment. The revised manuscript was checked by a native speaker again.

Round 2

Reviewer 2 Report

Comments to the Authors:

After my first revision, you have addressed successfully some of my comments, although there are still a few points left to be considered, as described below.

The statistical analysis is not clearly described yet. Which were the fixed and random effects in the statistical analysis? Why did you choose the Duncan multiple range test as the post-hoc test? You should explain how the specific comparisons were performed.

Some graphs (e.g., 1D, 1K, 2H and 2I) are still very difficult to be understood.

The results on the microbial composition are still confusing. In the presented figures it is not possible to see the significant difference in abundances (relative contributions) and the populations that apparently increased with RG or fRG and with Rd and protopanaxatriol (as described in the text, in lines 226-227 and 250-252) do not seem to appear in the supplemental Tables or the results presented there do not match with those explained in the text. As an example of the confusing presentation, Supplemental Table S4 is wrong because you use the symbol # for some populations in the Ec column and then you describe that symbol as a comparison with Ec. Again, the description of results on the microbial composition should be improved.

In the first line of the second paragraph of section 3.3 there is still an error because I assume you refer to the “oral gavage of Rd and protopanaxatriol” and not “of RG and fRG”. The same mistake appears in the legend of Figure 5.

Author Response

After my first revision, you have addressed successfully some of my comments, although there are still a few points left to be considered, as described below.

The statistical analysis is not clearly described yet. Which were the fixed and random effects in the statistical analysis? Why did you choose the Duncan multiple range test as the post-hoc test? You should explain how the specific comparisons were performed.

-->Thank you. We described the content related to your comment.

Some graphs (e.g., 1D, 1K, 2H and 2I) are still very difficult to be understood.

-->Thank you for your comment. We revised them in Figures.

The results on the microbial composition are still confusing. In the presented figures it is not possible to see the significant difference in abundances (relative contributions) and the populations that apparently increased with RG or fRG and with Rd and protopanaxatriol (as described in the text, in lines 226-227 and 250-252) do not seem to appear in the supplemental Tables or the results presented there do not match with those explained in the text. As an example of the confusing presentation, Supplemental Table S4 is wrong because you use the symbol # for some populations in the Ec column and then you describe that symbol as a comparison with Ec. Again, the description of results on the microbial composition should be improved.

-->Thank you. We revised them according to your comment.

In the first line of the second paragraph of section 3.3 there is still an error because I assume you refer to the “oral gavage of Rd and protopanaxatriol” and not “of RG and fRG”. The same mistake appears in the legend of Figure 5.

-->Thank you for your comment. We revised it.